# Risk Factors Associated with Intraoperative Iatrogenic Fracture in Patients Undergoing Intramedullary Nailing for Atypical Femoral Fractures with Marked Anterior and Lateral Bowing

**DOI:** 10.3390/medicina59040735

**Published:** 2023-04-09

**Authors:** Yong Bum Joo, Yoo Sun Jeon, Woo Yong Lee, Hyung Jin Chung

**Affiliations:** 1Department of Orthopedic Surgery, Chungnam National University Hospital, Chungnam National University College of Medicine, Daejeon 35015, Republic of Korea; 2Department of Orthopedic Surgery, Korea Worker’s Compensation & Welfare Service Daejeon Hospital, 637, Gyejok-ro, Daedeok-gu, Daejeon 34384, Republic of Korea; 3Department of Orthopedic Surgery, Chungnam National University Sejong Hospital, Chungnam National University College of Medicine, Sejong 30099, Republic of Korea

**Keywords:** diaphyseal atypical femoral fracture, intramedullary nail, iatrogenic fracture, femoral bowing angle, bone density

## Abstract

*Background and objectives*: Iatrogenic fractures are potential complications during intramedullary (IM) nailing for atypical femoral fractures (AFFs). The risk factors associated with iatrogenic fractures remain unclear, although excessive femoral bowing and osteoporosis are hypothesized to be contributing factors. The present study aimed to determine the risk factors for the occurrence of iatrogenic fractures during IM nailing in patients with AFFs. *Materials and Methods*: This retrospective cross-sectional study evaluated 95 patients with AFF (all female; age range: 49–87 years) who underwent IM nailing between June 2008 and December 2017. The patients were divided into two groups: Group I (with iatrogenic fracture: *n* = 20) and Group II (without iatrogenic fracture: *n* = 75). Background characteristics were retrieved from medical records and radiographic measurements were obtained. Univariate and multivariate logistic regression analyses were performed to identify risk factors for the occurrence of intraoperative iatrogenic fractures. Receiver operating curve (ROC) analysis was conducted to determine a cut-off value for the prediction of iatrogenic fracture occurrence. *Results*: Iatrogenic fractures occurred in 20 (21.1%) patients. The two groups exhibited no significant differences regarding age and other background characteristics. Group I exhibited significantly lower mean femoral bone mineral density (BMD) and significantly greater mean lateral and anterior femoral bowing angles than Group II (all *p* < 0.05). There were no significant differences in AFF location, nonunion, and IM nail diameter, length, or nail entry point between the two groups. In the univariate analysis, femoral BMD and lateral bowing of the femur differed significantly between the two groups. On multivariate analysis, only lateral bowing of the femur remained significantly associated with iatrogenic fracture occurrence. The ROC analysis determined a cut-off value of 9.3° in lateral bowing of the femur for prediction of iatrogenic fracture occurrence during IM nailing for AFF treatment. *Conclusions*: The lateral bowing angle of the femur is an important predictive factor for intraoperative iatrogenic fracture occurrence in patients undergoing IM nailing for AFF treatment.

## 1. Introduction

In 2013, according to the revised criteria of the American Society for Bone and Mineral Research (ASMBR) task force report, atypical femoral fractures (AFFs) are defined as “fractures located along the femoral diaphysis from just distal to the lesser trochanter to just proximal to the supracondylar flare”. In addition, at least four of the five major features must be present (Table 1) [1]. Recently, there has been increasing interest in AFFs, considered as a type of insufficiency fracture associated with long-term use of bisphosphonate (BP) [2]. BPs prevent osteoporotic fractures by inhibiting osteoclast-mediated bone resorption. However, the consequent decrease in bone turnover may compromise the mechanical and regenerative properties of the bone, resulting in fracture onset and delayed bone healing [2,3]. AFFs also occur in patients without exposure to BPs [1,4,5]. In the second report from the American Society of Bone and Mineral Research Task Force, the pathogenesis of AFFs was considered to involve stress or insufficiency [1]. Furthermore, lower limb geometry and Asian race may contribute to the risk of AFF occurrence [1,6].

The femur is one of the human bones that exhibits racial and sex differences [7]. Asian females have greater anterolateral bowing of the femur compared with White, African-American persons, and males, respectively [8]. With greater lateral bowing of the femur, AFFs are more likely to occur in the diaphyseal region than in the subtrochanteric region [9]. Moreover, in older patients, the occurrence of low energy diaphyseal femoral fracture can be attributed to an increased range of anterior and lateral bowing [10]. Therefore, curvature of the femur should be considered a potential contributor to AFF development.

Currently, intramedullary (IM) nailing is the preferred surgical method for AFF [1,11]. However, in cases with a mismatch between the bowing of the femur and the curvature of the IM nails, several problems can arise. Some challenges include iatrogenic fractures, straightening of the femur, medial gap opening, leg-length discrepancy, penetration of the distal anterior femoral cortex, delayed union, and nonunion [2,12,13,14,15,16,17]. Prasarn et al. [17] reported that the most frequent surgical complication in BP-associated AFFs was intraoperative cortical fracture during nail insertion. To overcome excessive bowing of the femur, some surgeons have examined the use of pre-bent nails according to the curvature of the femur or a nail for the opposite femur [15,16,18]. Park et al. [18] reported a new technique for IM nailing that relieves excessive bowing of the femur in AFFs by rotating the nail outward when the nail passes through the apex of the curve. Despite these methods, the possibility of an iatrogenic fracture during IM nailing remains a constant concern for surgeons [2,17,18,19]. When iatrogenic fracture occurs, it can cause instability of the inserted nail, which is one of the important problems for patients because it can affect bone healing. In addition, iatrogenic cortical fracture around the fracture site in complete AFFs was identified as an independent predictive factor for problematic healing [2].

To date, no studies have examined the risk factors associated with the occurrence of iatrogenic fractures during IM nailing in AFF treatment. The purpose of this study was to determine the risk factors that could lead to an iatrogenic fracture in patients undergoing IM nailing for AFFs. It was hypothesized that in AFFs, iatrogenic fractures during IM nailing are more likely to occur with excessive bowing of the femur and osteoporosis.

## 2. Material and Methods

### 2.1. Study Population

This was a retrospective cross-sectional study of 136 patients with AFF who underwent IM nailing between June 2008 and December 2017. The study was approved by our Institutional Review Board (Approval No. 2020-02-075, 19 March 2020), and the need for informed consent from all patients was waived. AFF was defined according to the second report of the American Society for Bone and Mineral Research (ASMBR) Task Force (Table 1) [1]. We examined the X-ray and computed tomography (CT) images to see whether there was periosteal or endosteal reaction of the lateral cortex thickening. Endosteal cortical thickening is defined as the increased cortical thickness of the fracture line just distal to the fracture site and formation of an endosteal callus, seen as “beaking” or “flaring” (Figure 1).

The following patients were excluded: (1) presence of comminuted fracture with high-energy trauma (*n* = 10); (2) bilateral AFFs (*n* = 9); (3) conservative treatment (*n* = 3); (4) internal fixation using proximal femoral nail antirotation or plate (*n* = 5); (5) use of glucocorticoid, proton pump inhibitor, or hormone (*n* = 3); (6) presence of metastatic bone tumor (*n* = 1); (7) unmeasurable bowing angle at the opposite intact femur (*n* = 2); and (8) no follow-up for more than 2 years after surgery (*n* = 7). Finally, 95 patients were included in the study. The patients were divided into two groups: Group I (with iatrogenic fracture; *n* = 20) and Group II (without iatrogenic fracture; *n* = 75) (Figure 2).

### 2.2. Surgical Procedures & Postoperative Managements

All operations were performed under aseptic conditions according to established surgical procedures. All patients were positioned supine on the fracture table, and then closed reduction was performed under fluoroscopy. Patients underwent surgical treatment with antegrade IM nail insertion. From June 2008 to November 2012, an IM nail (T2 Femoral Nailing System; Stryker, Schönkirchen, Germany) with an entry point located in the piriformis fossa was used (*n* = 16; 16.8%). From December 2012 to December 2017, a different IM nail (Expert Asian Femoral Nail, A2FN; Synthes, Solothurn, Switzerland) with an entry point located lateral to the tip of the greater trochanter was used (*n* = 79; 83.2%). The medullary cavity was reamed at 1.5 mm larger than the intended nail diameter. All surgeries were performed by two senior surgeons. After surgery, patients did not apply a long leg splint or brace. They were allowed to sit on the first postoperative day while wheelchair and partial weight bearing was initiated between the third and seventh postoperative days depending on the degree of reduction, systemic condition, and pain. Weight bearing was gradually increased according to the extent of fracture union determined by radiography.

### 2.3. Clinical Assessment

The medical records of the enrolled patients were reviewed to evaluate the demographic data. Data for surgical complication, BP intake and duration, smoking history, prodromal symptoms, body mass index (BMI), bone mineral density (BMD), Charlson comorbidity index (CCI) [20], American Society of Anesthesiologist (ASA) classification [21], and Koval walking grade [22] were collected. Dual-energy X-ray absorptiometry was performed to measure BMD in the anteroposterior direction of the lumbar spine and hip. In this way, we investigated the clinical differences between the two groups to identify the risk factors for iatrogenic fracture.

### 2.4. Radiographic Measurements

According to the fracture level, the AFFs were divided into proximal third, middle third, and distal third groups using the diaphyseal segment of the Arbeitsgemeinschaft für Osteosynthesefragen classification. An iatrogenic fracture was defined as an additional fracture lesion that newly occurred during surgery (Figure 3). Anteroposterior and lateral radiographs of the intact opposite femur were obtained to measure bowing of the femur. The bowing was defined as the angle between lines bisecting the femur at 0 and 5 cm below the lowest portion of the lesser trochanter and a line connecting points bisecting the femur at 5 and 10 cm above the distal articular surface [23] (Figure 4).

To investigate the effect of IM nail insertion on the occurrence of iatrogenic fracture, the entry points of the nails and the nail sizes were analyzed. The medullary cavity diameter of the fractured femur was measured at the narrowest shaft region on an anteroposterior radiograph of the femur. All patients were followed up for at least 2 years. At the last follow-up, fracture healing of the primary AFF and any iatrogenic fractures were evaluated. Fracture healing was defined as bridging of the fracture site by callus or bone at three cortices on plain radiographs [24]. All radiologic evaluations were conducted by two of the authors using plain radiographs of the femur with a picture archiving and communication system workstation (Maroview, version 5.4.10.52; Marotech, Seoul, Republic of Korea). Measurements were performed twice by both authors and the average values were used. To eliminate memory effects, the measurements were conducted 4 weeks apart. The authors performed the measurements independently without information about the patients. Intraclass correlation coefficients (ICCs) were assessed to determine inter- and intra-observer agreements for the radiologic measurements. The ICCs for agreement were interpreted as follows [25]: <0.5: poor; ≥0.5 but ≤0.9: good; and >0.9: excellent. The inter- and intra-observer reliabilities for all radiologic measurements were excellent (ICC > 0.90). As such, we identified secondary outcomes through radiologic measurement.

### 2.5. Statistical Analysis

Statistical analyses were conducted using SPSS software (version 26.0; IBM Corporation, Armonk, NY, USA). Univariate analysis was used to identify significance between-group differences, defined as *p* < 0.05. Fisher’s exact test was performed when the number of categorical variables was 2, and a chi-square test was performed when the number of categorical variables was ≥3. The frequency and percentage of the categorical variables were presented. Student’s *t*-test was used to analyze continuous variables. When heterogeneity of variance was found in the distribution of the continuous variables using Levene’s test, Welch’s method was applied. Statistical data were presented as means ± standard deviations. Univariate and multivariate logistic regression analyses were performed to identify risk factors for an iatrogenic fracture as the primary outcome. Odds ratios (ORs) and 95% confidence intervals (CIs) were calculated for relative risks. In the multivariate analysis, a receiver operating characteristic (ROC) curve analysis was conducted for variables that were significantly related to predict iatrogenic fracture occurrence. The ROC curve analysis was also used to identify a cut-off value for lateral bowing angle of the femur to predict iatrogenic fracture. The cut-off point on the ROC curve is equivalent to the point at which the sensitivity and specificity were maximal as a secondary outcome.

## 3. Results

All 95 patients were female and iatrogenic fractures occurred in 20 (21.1%). There were no significant differences in age, affected femur (left and right sides), BMI, smoking habit, duration of BPs, CCI, ASA classification, Koval score, and prodromal symptoms between the two groups (Table 2).

The fracture characteristics are shown in Table 3. The mean femoral BMD in Group I (−2.9; range: −4.3 to −0.8) was significantly lower than that in Group II (−2.5; range: −4.5 to −0.3; *p* = 0.046). The mean lateral bowing angle in Group I (14.7° ± 5.9°; range: 2.4–24.8°) was significantly higher than that in Group II (7.9° ± 6.5°; range: 0.2–21.9°; *p* < 0.001). The mean anterior bowing angle in Group I (16.6° ± 4.2°; range: 9.7–25.1°) was also significantly higher than that in Group II (13.0° ± 7.8°; range: 1.0–29.5°; *p* = 0.008). However, no significant differences were observed in spinal BMD, diaphyseal AFF location, nail entry point, nail diameter, nail length, medullary cavity diameter and the difference (ΔD) in mm between the inner canal and IM nail diameter between the two groups. Nonunion occurred in 1 of 20 patients (5.0%) in Group I, and 5 patients (6.7%) in Group II (*p* = 1.000). All iatrogenic fracture sites were completely healed (Figure 5). The time to full weight bearing after surgery in Group I (24.4 ± 1.3 days; range: 20–28) was significantly longer than that in Group II (1.6 ± 0.3 day; range: 1–5; *p* < 0.001).

The results of univariate and multivariate logistic regression analyses are shown in Table 4. In the univariate analyses, femoral BMD and lateral bowing of the femur exhibited significant between-group differences (*p* = 0.050 and *p* < 0.001, respectively). In the multivariate analysis, a significant association was identified between lateral bowing of the femur and iatrogenic fracture (adjusted OR = 1.205; 95% CI: 1.046–1.389; *p* = 0.010), whereas femoral BMD was not identified as a significant risk factor.

In the ROC curve analysis, the area under the curve for lateral bowing of the femur and iatrogenic fracture was significant (0.786; *p* = 0.010) and the cut-off value for lateral bowing of the femur to predict iatrogenic fracture was 9.3° (Figure 6).

## 4. Discussion

In this study, we confirmed that greater lateral bowing of the femur is an important predictive factor for intraoperative iatrogenic fracture in patients undergoing IM nailing for AFFs.

AFFs are often associated with long-term use of BPs [26,27,28,29,30] but can also occur in patients without BP use, and increased femoral curvature may be an important causative factor for low-energy AFFs [1,9,10]. AFFs are stress fractures and that the geometry of the entire lower extremity can contribute to altered stress on the lateral cortex of the femur [1,6,9]. Anterolateral bowing of the femur is a risk factor for AFFs and the risk is five times higher in Asian populations compared with Caucasian populations [6,8,18]. In addition, lateral bowing of the femur was shown to determine the location of AFFs [3,9]. Yoo et al. [3] reported that the mean lateral bowing of the femur was 10.1° ± 3.79° in the diaphyseal AFF group and 3.3° ± 2.4° in the subtrochanteric AFF group. When the lateral bowing of the femur was greater than 5.2°, diaphyseal AFFs were more frequent than subtrochanteric AFFs. Kim et al. [9] also reported that the mean lateral bowing in the diaphyseal AFF group was significantly greater than that in the subtrochanteric AFF group (7.8° ± 4.8° versus 1.6° ± 1.8°). In other words, they found that as the lateral bowing of the femur increased, AFFs tended to occur more often in the diaphyseal region than in the subtrochanteric region. In the present study, the mean lateral bowing of the femur was 14.7° ± 5.9° in Group I (with iatrogenic fracture) and 7.9° ± 6.5° in Group II (without iatrogenic fracture). Similar to the findings in previous studies, all patients displayed large lateral bowing of the femur. In particular, the lateral bowing of the femur in Group I was twice as large as that in Group II, which was a meaningful finding related to iatrogenic fracture occurrence (Table 2).

Mismatching between the bowing of the femur and the curvature of the IM nail can produce an eccentric position of the distal nail tip and lead to an iatrogenic fracture due to straightening of the femur [13,14,17]. This complication can also lead to medial gap opening, leg-length discrepancy, and nonunion [2,16]. The incidence of iatrogenic fracture during IM nailing in AFFs currently remains unclear. Lim et al. [2] reported an incidence of 4.6% (5/109) whereas Prasarn et al. [17] reported 29.4% (5/17). The incidence in the present study was 21.1% (20/95), which is in agreement with those of previous studies. However, the incidence appears to vary from study to study because the participants are different and there are slight differences in surgical techniques. Therefore, more research is required to clarify the incidence. The present study focused on determining risk factors for iatrogenic fracture occurrence during IM nailing in AFFs. In our study, Group I displayed greater lateral and anterior bowing of the femur than Group II (Table 2). Therefore, because of the large curvature of the femur, there was considerable nail mismatching in Group I and iatrogenic fractures were more likely to occur. However, anterior bowing of the femur was excluded as a risk factor for iatrogenic fracture occurrence in the multivariate analysis (Table 3). Therefore, we confirmed that greater lateral bowing of the femur was a significant risk factor for iatrogenic fracture occurrence during IM nailing for AFFs.

Increased bowing of the femur is related to impaired fracture healing in AFFs, and lateral bowing of the femur exceeding 10° may contribute to the high rate of delayed union [2]. Meanwhile, iatrogenic cortical fracture around the primary fracture site displayed problematic healing with an adjusted OR of 19.7 in complete AFFs [2]. Therefore, if an iatrogenic fracture occurs during IM nailing in AFFs with excessive lateral bowing of the femur, the risk of complications such as nonunion or delayed union may be increased. In our cohort, there was no between-group difference in the healing rate, although Group I exhibited excessive lateral bowing (14.7°) and additional iatrogenic fractures. The difference in results compared to other studies may potentially be due to the different patient cohorts and surgical procedures. Our results further showed that even if an iatrogenic fracture occurred during IM nailing, there was no effect on final bone union. However, for patients with an iatrogenic fracture, weight-bearing may begin later or rehabilitation may take longer, even if there is no effect on fracture healing. This can lead to increased mortality and morbidity [31]. In the present study, the patients in Group II were allowed to undertake immediate postoperative full weight bearing, whereas the patients in Group I began full weight bearing at a mean of 24.4 days after surgery (*p* < 0.001) (Table 2). Therefore, for early rehabilitation, it is important to prevent iatrogenic fracture occurrence during surgery. Although we cannot completely overcome the risk of iatrogenic fracture occurrence during IM nailing in AFFs in patients with excessive bowing of the femur, we can predict its likelihood and carefully select an appropriate implant before performing the procedure.

AFFs represent a form of osteoporotic fracture [32]. AFFs may be related to age >65 years, more lateral bowing, Asian females, and lower BMD [8,9,33,34,35]. Unexpected iatrogenic fractures can occur during IM nailing in the diaphyseal region of the femur in patients with osteoporosis [36]. Therefore, as the severity of osteoporosis increases, the probability of developing an iatrogenic fracture during IM nailing in AFFs becomes higher. In the present study, the mean femoral BMD in Group I was significantly lower than that in Group II (Table 2). However, femoral BMD was not a significant risk factor in the multivariate analysis (Table 3). Nevertheless, decreased femoral BMD should not be completely ignored as a risk factor because osteoporotic bone is susceptible to fracture [37].

To our knowledge, this is the first study to determine the risk factors for intraoperative iatrogenic fracture occurrence during intramedullary nailing for AFFs. The strength of the study is that factors that can cause iatrogenic fractures were evaluated using univariate and multivariate logistic regression analyses. The univariate analyses revealed correlations with iatrogenic fractures for femoral BMD, as well as lateral and anterior bowing of the femur. The multivariate analysis confirmed that lateral bowing of the femur was a reliable risk factor associated with iatrogenic fracture occurrence. In addition, the cut-off value for lateral bowing of the femur with significant sensitivity and specificity for prediction of iatrogenic fracture occurrence was determined at 9.3°. In other words, iatrogenic femoral fracture occurrence is more frequent during IM nailing for AFFs when the lateral bowing of the femur is larger than 9.3°. Thus, when IM nailing is planned for AFF, measurement of the lateral bowing of the femur should be considered in the preoperative planning to provide a reference value for predicting iatrogenic fracture occurrence.

According to previous studies, various methods have been proposed to reduce the incidence of iatrogenic fractures during IM nailing. For example, the entry point of the nail was planned with caution in order to avoid anterior or external deviation. Proper reaming for bowing shape was also selected [38,39]. In 2017, Park et al. reported new grading systems for anterolateral femoral bowing [18]. Furthermore, they introduced the new intramedullary nailing technique by which the nail is rotated externally for femur bowing. Recently, three-dimensional (3D) technologies have been introduced to help prevent iatrogenic fractures by planning surgery in advance [40].

In this way, various methods have been introduced to prevent iatrogenic fracture during surgery, and we also contemplated means to prevent iatrogenic fractures based on these factors. By extension, we considered the risk factors for iatrogenic fractures and identified significant risk factors through the cut-off value of the lateral bowing angle. Based on this, surgeons may benefit from identifying the bowing angle of the femur before surgery to evaluate the predictability of perioperative iatrogenic fractures and make efforts to reduce iatrogenic fractures.

The present study has several limitations. First, because the study was retrospective, bone metabolic markers were not evaluated before the patients presented with fractures. Second, the study was not a randomized controlled trial. A larger, multicenter, prospective study is required to confirm our findings. Third, because the lateral and anterior bowing angles were measured on the opposite intact femur, these angles could be different from those of the fractured femur. Fourth, the study was limited to Asians. Therefore, various races need to be evaluated in prospective studies to confirm the present findings.

## 5. Conclusions

The present study analyzed the risk factors for intraoperative iatrogenic fracture during IM nailing for diaphyseal AFFs. Lateral bowing of the femur was identified as a significant risk factor and its cut-off value for prediction of an intraoperative iatrogenic fracture was 9.3°. Surgeons should evaluate the lateral bowing of the femur during preoperative planning for diaphyseal AFFs, and if patients exhibit large lateral bowing of the femur, care should be taken to prevent the occurrence of iatrogenic fractures during surgery.

## Figures and Tables

**Figure 1 medicina-59-00735-f001:**
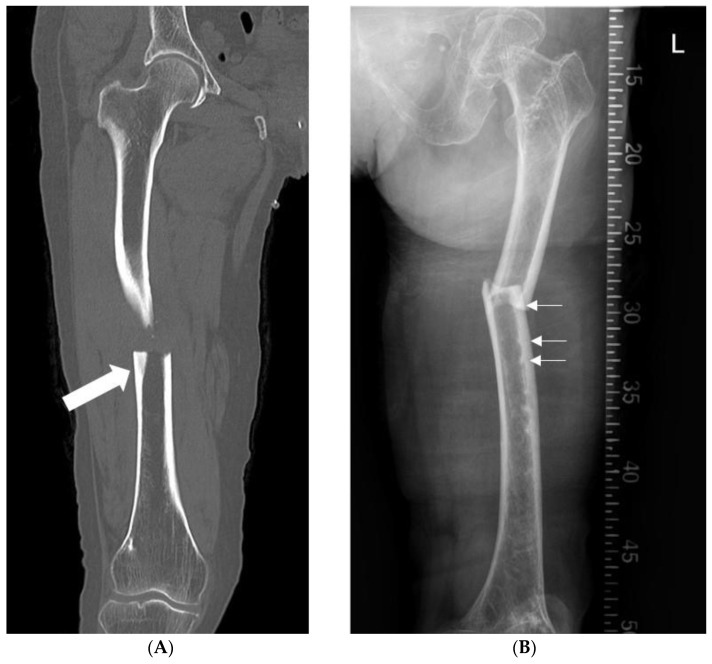
(**A**) Atypical femur fracture in the right femur CT view. Lateral cortical thickening at the fracture site can be observed (depicted by block arrow). (**B**) Atypical femur fracture in the left femur scanogram. Endosteal cortical thickening can be observed at multiple sites near the fracture lesion (depicted by line arrows).

**Figure 2 medicina-59-00735-f002:**
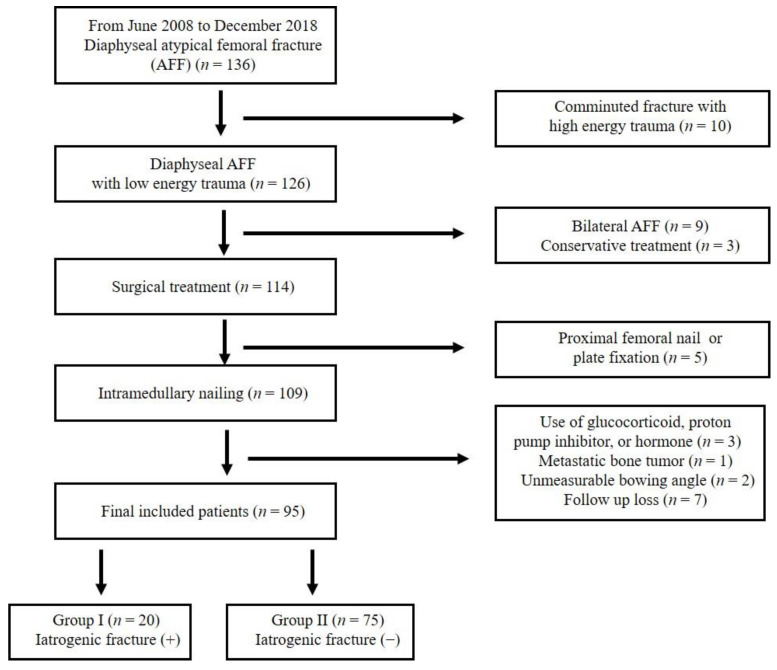
Flow chart of the patient selection.

**Figure 3 medicina-59-00735-f003:**
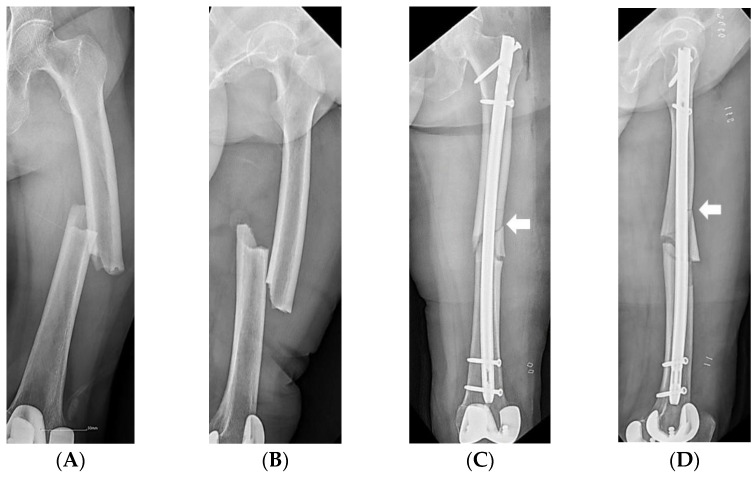
(**A**,**B**) X-rays of a 74-year-old woman showing a complete diaphyseal atypical femoral fracture. (**C**,**D**) Postoperative radiographs showing the intramedullary nail fixation and an iatrogenic fracture (arrows).

**Figure 4 medicina-59-00735-f004:**
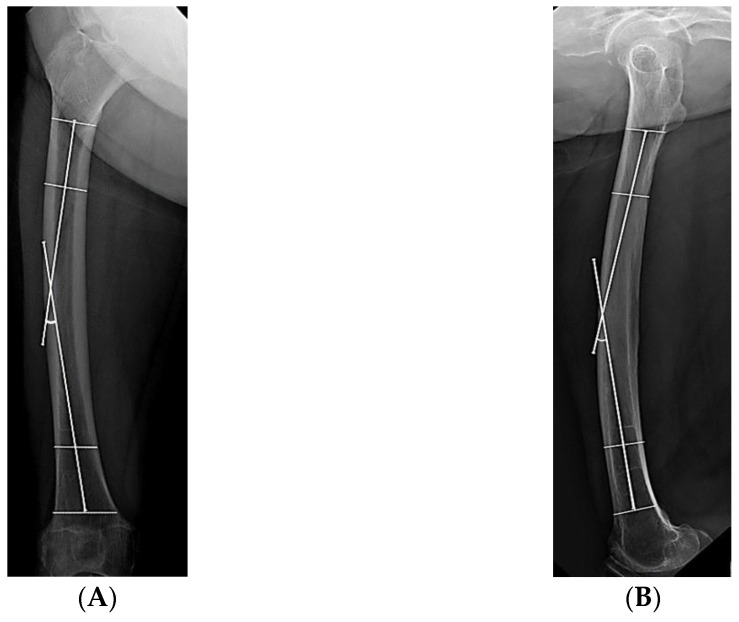
Measurement of femoral bowing in the opposite intact femur. (**A**) Lateral bowing on an anteroposterior X-ray. (**B**) Anterior bowing on a lateral X-ray.

**Figure 5 medicina-59-00735-f005:**
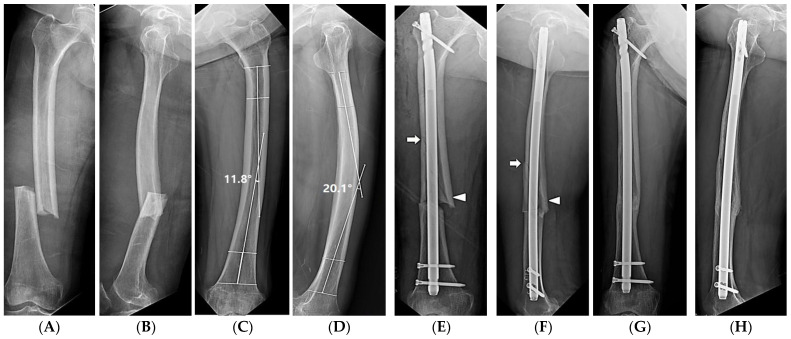
A series of X-rays from a 79-year-old female patient who underwent intramedullary nailing under the diagnosis of AFF and experienced an iatrogenic diaphyseal fracture. (**A**,**B**) X-ray images of the AFF from the first visit to the emergency room. (**C**,**D**) X-ray images of the opposite intact femur. A lateral bowing angle of 11.8° and an anterior bowing angle of 20.1° were measured. (**E**,**F**) Medial gap opening (arrowheads) and iatrogenic diaphyseal fracture (arrows) occurred during intramedullary nailing. (**G**,**H**) Osseous union was obtained at 2 years after surgery.

**Figure 6 medicina-59-00735-f006:**
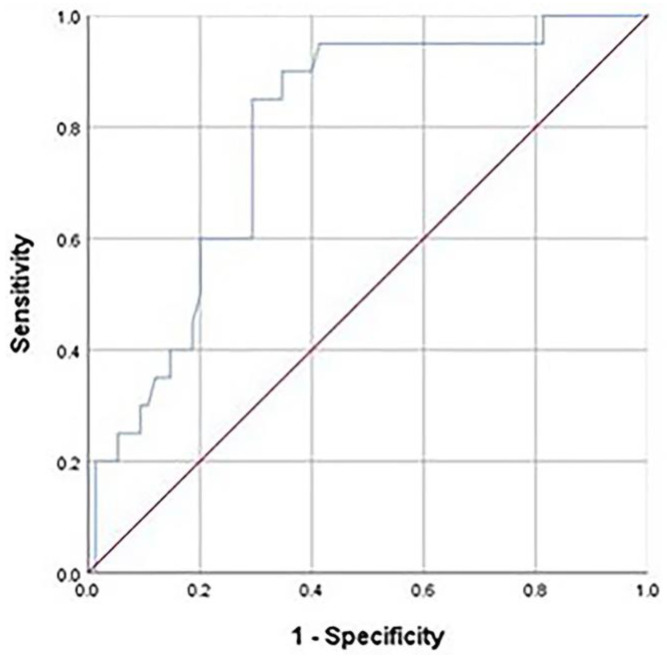
Receiver operating characteristic curve of the lateral bowing angle. The area under the curve, cut-off value, Youden’s index, sensitivity, and specificity of the lateral bowing angle were 0.786 (0.684–0.888), 9.30, 0.557, 0.850, and 0.707, respectively (*p* < 0.001).

**Table 1 medicina-59-00735-t001:** ASBMR Task Force 2013 revised case definition of AFFs.

Major Criteria
✓ The fracture is associated with minimal or no trauma, as in a fall from a standing height or less. ✓ The fracture line originates at the lateral cortex and is substantially transverse in its orientation, although it may become oblique as it progresses medially across the femur. ✓ Complete fractures extend through both cortices and may be associated with a medial spike; incomplete fractures involve only the lateral cortex. ✓ The fracture is noncomminuted or minimally. ✓ Localized periosteal or endosteal thickening of the lateral cortex is present at the fracture site (‘beaking’ or ‘flaring’).
**Minor Criteria**
✓ Generalized increase in cortical thickness of the femoral diaphyses ✓ Unilateral or bilateral prodromal symptoms such as dull or aching pain in the groin or thigh ✓ Bilateral incomplete or complete femoral diaphysis fractures ✓ Delayed fracture healing

**Table 2 medicina-59-00735-t002:** Comparison of demographic data between patients with and without iatrogenic fracture.

Variables	Group I (Iatrogenic Fracture, *n* = 20)	Group II (Non-Iatrogenic Fracture, *n* = 75)	*p*-Value
Age, years	73.5 ± 10.9 (50–87)	71.8 ± 9.0 (49–86)	0.472
Female sex	20 (100%)	75 (100%)	1.000
Affected side, Rt: Lt	11 (55.0%):9 (45.0%)	32 (42.7%):42 (57.3%)	0.449
Height, cm	150.3 ± 6.1 (138–160)	151.4 ± 7.2 (125–165)	0.550
Weight, kg	52.9 ± 8.6 (38–73)	53.8 ± 8.1 (35–75)	0.662
Body mass index, kg/m^2^	23.4 ± 3.4 (18.1–32.9)	23.5 ± 3.3 (16.7–34.9)	0.893
Smoking, *n*	0 (0.0%)	4 (5.3%)	0.576
Bisphosphonate use	14 (70%)	61 (81%)	0.354
Duration of BPs, month	59.1 ± 7.1	66.5 ± 4.0	0.225
Charlson comorbidity index			0.739
1 or 2	1 (5.0%)	6 (8.0%)	
3 or 4	12 (60.0%)	46 (61.3%)	
≥5	7 (35.0%)	23 (30.7%)	
ASA ^a^ classification			0.158
I or II	19 (95.0%)	54 (72.0%)	
III or IV	1 (5.0%)	21 (28.0%)	
Koval score			0.359
1	19 (95.0%)	61 (81.3%)	
2 or 3	0 (0.0%)	11 (14.7%)	
4 or 5	0 (0.0%)	2 (2.7%)	
6 or 7	1 (5.0%)	1 (1.3%)	
Prodromal symptom	2 (10.0%)	16 (21.3%)	0.345

Data are expressed as mean ± standard deviation or number (percentage). ^a^, ASA American Society of Anesthesiologists.

**Table 3 medicina-59-00735-t003:** Comparison of fracture- and surgery-related characteristics between patients with and without iatrogenic fracture.

Variables	Group I (Iatrogenic Fracture, *n* = 20)	Group II (Non-Iatrogenic Fracture, *n* = 75)	*p*-Value
Spinal BMD ^a^, T-score	−3.1 ± 1.1 (−5.2 to −1.1)	−2.8 ± 0.8 (−4.8 to −0.9)	0.204
Femoral BMD, T-score	−2.9 ± 1.0 (−4.3 to −0.8)	−2.5 ± 0.9 (−4.5 to −0.3)	0.046 ^b^
AFF ^c^ location			0.803
Proximal third, *n*	6 (30.0%)	18 (24.0%)	
Middle third, *n*	13 (65.0%)	51 (68.0%)	
Distal third, *n*	1 (5.0%)	6 (8.0%)	
Lateral bowing, °	14.7 ± 5.9 (2.4–24.8)	7.9 ± 6.5 (0.2–21.9)	<0.001 ^b^
Anterior bowing, °	16.6 ± 4.2 (9.7–25.1)	13.0 ± 7.8 (1.0–29.5)	0.008 ^b^
Nail entry point Greater trochanter, *n*	17 (85.0%)	62 (82.7%)	1.000
Piriformis, *n*	3 (15.0%)	13 (17.3%)	1.000
Nail diameter, mm	11.9 ± 1.2 (9–13)	11.5 ± 1.4 (9–14)	0.185
Nail length, mm	334.0 ± 27.6 (280–380)	336.0 ± 23.5 (280–380)	0.745
Medullary cavity diameter, mm	13.3 ± 2.0 (9.6–16.6)	12.8 ± 2.3 (8.2–17.8)	0.328
Medullary cavity—Nail diameter, mm			0.125
<1 mm, *n*	8 (40.0%)	37 (49.3%)	
1–2 mm, *n*	8 (40.0%)	14 (18.7%)	
>2 mm, *n*	4 (20.0%)	24 (32.0%)	
Nonunion, *n*	1 (5.0%)	5 (6.7%)	1.000
Full-weight bearing after surgery, day	24.4 ± 1.3 (20–28)	1.6 ± 0.3 (1–5)	<0.001 ^b^
Final follow up period, months	29.3 ± 8.3 (24–59)	31.9 ± 10.2 (24–73)	0.294

Data are expressed as mean ± standard deviation (range) or number (percentage). ^a^ BMD bone mineral density, ^b^ Statistically significant, ^c^ AFF Atypical femoral fracture.

**Table 4 medicina-59-00735-t004:** Univariate and multivariate analysis of variables associated with iatrogenic fracture.

Variables	Crude OR (95% CI)	*p* Value	Adjusted OR (95% CI)	*p* Value
Spine BMD ^a^	0.701 (0.406–1.213)	0.205	1.185 (0.441–3.183)	0.737
Femoral BMD	0.546 (0.299–0.999)	0.050 ^b^	0.577 (0.208–1.605)	0.292
Lateral bowing, °	1.154 (1.065–1.251)	<0.001 ^b^	1.205 (1.046–1.389)	0.010 ^b^
Anterior bowing, °	1.067 (0.997–1.141)	0.061	1.048 (0.915–1.199)	0.500

^a^ BMD bone mineral density, ^b^ Statistically significant.

## Data Availability

The data presented in this study are available on request from the corresponding author.

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
