# Peer review of "Risk Factors Associated with Intraoperative Iatrogenic Fracture in Patients Undergoing Intramedullary Nailing for Atypical Femoral Fractures with Marked Anterior and Lateral Bowing"

_medicina, 2023, doi:10.3390/medicina59040735_

Round 1
Reviewer 1 Report
Thank you for the opportunity to review a paper entitled " Risk factors associated with intraoperative iatrogenic fracture 2 in patients undergoing intramedullary nailing for atypical femoral fractures ". Fracture treatment for AFF is challenging and has a higher rate of complications than common femoral fractures. This study evaluated risk factors for iatrogenic fractures in 95 cases of AFF. As a result, lateral bowing was extracted as an independent risk factor, and the result of the ROC curve yielded a cutoff value of 9.3 degrees. I have some specific remarks as following.
・Introduction; I feel that the outline and problems of AFF are described briefly. I think it would be better to emphasize what kind of demerit the patient suffers from iatrogenic fracture in AFF.
・Method; Readers would like to know the evaluation of the presence or absence of bone union at the fracture site and the timing of bone union. Were there any cases of additional surgery or delayed union?
・Method; It is necessary to describe how post-treatment is performed.
・Method; How do you take bisphosphonate use after surgery?.
・Method; In AFF, the skill of the operator greatly affects the results, but how many operators are involved in this research? Also, were there any differences in iatrogenic fractures in terms of the skill of the surgeon?
・Method; A power analysis should be performed to assess whether a statistically valid number of cases has been obtained.
・Results; The results are briefly described.
・Discussion; The authors state that the occurrence of iatrogenic fractures in AFF can lead to complications. On the other hand, this study did not present any results regarding this point. It is necessary to evaluate whether there were any problems with bone union and the time required for bone union in Group I.
・Methods; It would be better to add what kind of ingenuity is necessary for cases with strong lateral bowing. Based on these results, how do the authors now deal with cases with strong lateral bowing?
・Table & Figure; Tables and figures feel easy to understand and described.
Author Response
Author response to reviewer’s comments
Thank you so much for reviewing this paper. We are honored to be given the opportunity to revise and polish the paper once again. We have done our best to revise the paper as per the reviewer‘s comments in order to advance our paper further. We have made the following revisions as per your comments:
Reviewer #1
・Introduction; I feel that the outline and problems of AFF are described briefly. I think it would be better to emphasize what kind of demerit the patient suffers from iatrogenic fracture in AFF.
- Thank you for your comment. I agree with your opinion.
- As you mentioned, I have revised the context to emphasize the patient suffers from iatrogenic fracture in AFF.
- Line 68-70
・Method; Readers would like to know the evaluation of the presence or absence of bone union at the fracture site and the timing of bone union. Were there any cases of additional surgery or delayed union?
- Thank you for your comment.
- Nonunion occurred in 1 of 20 patients (5.0%) in Group I, and 5 patients (6.7%) in Group II (p = 1.000). But, all iatrogenic fracture sites were completely healed. There was no additional surgery for delayed union. And, there was no significant difference for union period in each group.
- Line 208-212
・Method; It is necessary to describe how post-treatment is performed.
- Thank you for your comment.
- The manuscript has been revised accordingly.
- Line 105, 115-120
・Method; How do you take bisphosphonate use after surgery?.
- In our study, patients taking BP were discontinued after surgery and injection with teriparatide was replaced. As you know, The current study demonstrated that the radiographic union rate by six months after treatment was significantly higher in the teriparatide group than in the bisphosphonate group. However, it was not possible to apply it to all patients after surgery due to problems such as cost.
-Method; In AFF, the skill of the operator greatly affects the results, but how many operators are involved in this research? Also, were there any differences in iatrogenic fractures in terms of the skill of the surgeon?
- Thank you for your comment. I agree with your opinion.
- As noted in the manuscript (Line 115), All surgeries were performed by two senior surgeons. There was no difference in the incidence of iatrogenic fractures between the two operators.
・Method; A power analysis should be performed to assess whether a statistically valid number of cases has been obtained.
- Thank you for your comment. We tried to investigate every available variable, including demographics and comorbidities that had been known to be associated with AFF. Univariate analysis was used to identify significant between-group differences. Fisher’s exact test was performed when the number of categorical variables was 2, and a chi-square test was performed when the number of categorical variables was ≥3. The frequency and percentage of the categorical variables were presented. Student’s t-test was used to analyze continuous variables. When heterogeneity of variance was found in the distribution of the continuous variables using Levene’s test, Welch’s method was applied.
・Results; The results are briefly described.
- Thank you for your comment.
・Discussion; The authors state that the occurrence of iatrogenic fractures in AFF can lead to complications. On the other hand, this study did not present any results regarding this point. It is necessary to evaluate whether there were any problems with bone union and the time required for bone union in Group I.
- Thank you for your comment.
- As noted in the manuscript (Line 209-213, Table 2&3), Nonunion occurred in 1 of 20 patients (5.0%) in Group I, and 5 patients (6.7%) in Group II. Actually, as you know, it was ambiguous to determine the exact time of bone union in each group given that most patients revisit the outpatient clinics in 2 to 3 months’ interval postoperatively. For this reason, we did not express it as “the bone union period”, but instead used the term “the final follow-up period.” (If it was considered that the bone union was complete, outpatient follow up was complete as well.)
- All iatrogenic fracture sites were completely healed in group I, the time to full weight bearing after surgery in Group I (24.4 ±1.3 days; range: 20–28) was significantly longer than that in Group II (1.6 ± 0.3 day; range: 1–5; p< 0.001).
・Methods; It would be better to add what kind of ingenuity is necessary for cases with strong lateral bowing. Based on these results, how do the authors now deal with cases with strong lateral bowing?
- Thank you for your considerable recommendation. I agree with your opinion.
- In previous studies, various methods have been proposed to reduce the incidence of iatrogenic fractures during IM nailing. In this study, we did not focus on surgical techniques, but tried to identify risk factors. Therefore, we thought that our finding (cut-off value of bowing angle) has the advantages for avoiding iatrogenic fractures.
- We described this and the manuscript has been revised accordingly.
・Table & Figure; Tables and figures feel easy to understand and described.
- Thank you for your comment.

Reviewer 2 Report
I think the main aim of the study is lateral bowing as a factor in iatrogenic fracture.
That need to be clear in the title
lateral bowing is not easy to assess in plan X ray, that make the methods of assessment used in this paper not sensitive enough
Author Response
Author response to reviewer’s comments
Thank you so much for reviewing this paper. We are honored to be given the opportunity to revise and polish the paper once again. We have done our best to revise the paper as per the reviewer‘s comments in order to advance our paper further. We have made the following revisions as per your comments:
Reviewer #2
I think the main aim of the study is lateral bowing as a factor in iatrogenic fracture.
That need to be clear in the title
- Thank you for your comment. I agree with your opinion.
- As you mentioned, I have revised the title to emphasize the femoral bowing,
lateral bowing is not easy to assess in plan X ray, that make the methods of assessment used in this paper not sensitive enough
- Thank you for your comment.
- Unfortunately, we were not able to perform MRI scanning on AFFs patients, but we did thoroughly check their x-ray and CT images. As you mentioned, it is not easy to measure the lateral bowing angle exactly in x-ray. However, by using the the x-ray & CT images, we could measure the bowing angle indirectly. Although they were not absolute values, they are thought to be meaningful as relative values to compare with each other.

Reviewer 3 Report
The paper presents a retrospective study aimed to determine the risk factors for the occurrence of iatrogenic fractures during intramedullary nailing in patients with atypical femoral fractures.
After the title, the article should present the names of the authors, their institutional affiliations and contact details.
The work is presented in each paragraph with great clarity and accuracy.
The summary is structured according to the content of the work.
The introduction transposes the topic of the research and finally presents the objective of the research.
The research methodology is clearly structured and the statistical methods are described.
The results are presented synthetically and their discussion is related to other achievements in the field.
At the end of the discussion paragraph, along with the study direction regarding the evaluation of different races in prospective studies to confirm the findings of the current studies, the use of preoperative planning could also be included, such as:
Moldovan, F.; Gligor, A.; Bataga, T. Structured Integration and Alignment Algorithm: A Tool for Personalized Surgical Treatment of Tibial Plateau Fractures. J. Pers. Med.2021, 11, 190. https://doi.org/10.3390/jpm11030190.
This would expand and diversify the bibliographic list.
The bibliographic references are correctly written.
There are some editing errors: A space should be left between the text and the citation resource, e.g.: "present (Table 1)[1].", but also the following indicated references.
I congratulate the authors for the research carried out.
Author Response
Author response to reviewer’s comments
Thank you so much for reviewing this paper. We are honored to be given the opportunity to revise and polish the paper once again. We have done our best to revise the paper as per the reviewer‘s comments in order to advance our paper further. We have made the following revisions as per your comment
Reviewer #3
The paper presents a retrospective study aimed to determine the risk factors for the occurrence of iatrogenic fractures during intramedullary nailing in patients with atypical femoral fractures.
After the title, the article should present the names of the authors, their institutional affiliations and contact details.
- Thank you for your comment. The manuscript has been accordingly.
The work is presented in each paragraph with great clarity and accuracy.
The summary is structured according to the content of the work.
The introduction transposes the topic of the research and finally presents the objective of the research.
The research methodology is clearly structured and the statistical methods are described.
The results are presented synthetically and their discussion is related to other achievements in the field.
- Thank you for your comment.
At the end of the discussion paragraph, along with the study direction regarding the evaluation of different races in prospective studies to confirm the findings of the current studies, the use of preoperative planning could also be included, such as:
Moldovan, F.; Gligor, A.; Bataga, T. Structured Integration and Alignment Algorithm: A Tool for Personalized Surgical Treatment of Tibial Plateau Fractures. J. Pers. Med.2021, 11, 190. https://doi.org/10.3390/jpm11030190.
- Thank you for your considerable recommendation.
- As you mentioned, I have revised the end of the discussion paragraph.
This would expand and diversify the bibliographic list.
The bibliographic references are correctly written.
There are some editing errors: A space should be left between the text and the citation resource, e.g.: "present (Table 1)[1].", but also the following indicated references.
- Thank you for your comment. The manuscript has been revised accordingly.
I congratulate the authors for the research carried out.
- Thank you for your comment.

Round 2
Reviewer 2 Report
the paper now is OK